# Adaptation of the Protocol for the Isolation of Biotinylated Protein Complexes for *Drosophila melanogaster* Tissues

**DOI:** 10.3390/ijms26168009

**Published:** 2025-08-19

**Authors:** Igor A. Shokodko, Rustam H. Ziganshin, Nadezhda E. Vorobyeva

**Affiliations:** 1Institute of Gene Biology Russian Academy of Sciences, 119334 Moscow, Russia; 2Shemyakin-Ovchinnikov Institute of Bioorganic Chemistry Russian Academy of Sciences, 117997 Moscow, Russia

**Keywords:** proximal labeling, proximal biotinylation, TurboID, ecdysone, EcR, biotin, ecdysone receptor, streptavidin, modified streptavidin, mass spectrometry, LC-MS/MS, transcription regulation

## Abstract

Proximity biotinylation, which utilizes various biotin ligating enzymes (BioID, TurboID, etc.), is widely used as a powerful tool for identifying novel protein–protein interactions. However, this method has a significant limitation: the use of streptavidin on beads for enriching biotinylated proteins often results in a high background of peptides from streptavidin itself, which interferes with identification by peptide mass fingerprinting. This limitation makes it practically impossible to study samples containing a small amount of material, such as individual insect tissues. In this study, we compared different precipitation and elution conditions for the purification of biotinylated proteins from protein extracts of *Drosophila melanogaster* S2 cells. We found that biotinylated proteins can be purified using anti-biotin antibodies, although with lower efficiency than streptavidin-based resin. We also demonstrated that protease-resistant streptavidin (prS), previously tested in mammalian cells, can be used effectively to purify biotinylated proteins from tissues of *D. melanogaster*. In our experiments, prS showed precipitation efficiency comparable to regular streptavidin but generated a lower background in peptide fingerprinting. To further demonstrate the applicability of prS for studying protein–protein interactions in *D. melanogaster* tissues, we carried out experiments to identify interaction partners of the ecdysone receptor (EcR) in *D. melanogaster* ovarian tissue using TurboID-based proximity biotinylation. As a result, EcR was found to interact with both previously described and novel protein partners in *Drosophila* ovaries.

## 1. Introduction

*Drosophila melanogaster* serves as an effective model system for studying human pathologies. A widely used approach is to investigate the role of newly identified human disease genes in *Drosophila* physiological processes [1,2]. Approximately 75% of genes implicated in human diseases have functional homologs in fruit flies, providing a powerful platform for examining conserved pathological mechanisms [3]. The extensive databases of *Drosophila* mutant stocks further enhance their status as a genetic toolkit. However, biochemical studies of protein–protein interactions between target proteins in *Drosophila* remain challenging due to the limited quantity of biological material available. Consequently, more sensitive methods—such as proximity biotinylation (PB)—are being adopted in place of traditional immunochemical techniques. PB methods (including BioID, APEX, TurboID, and others) offer significant advantages: they enable the identification of even labile and low-affinity interactions, and they are effective when only minimal amounts of material are available [4].

To identify protein partners using PB, the target protein is typically expressed as a chimera fused to a biotin ligase. The activated biotin ligase catalyzes the transfer of biotin (either exogenous or endogenous) to amino groups of proteins that are in close proximity to the target protein [5]. The resulting biotinylated proteins can then be purified using streptavidin-based resins, enabling the identification of target protein partners by mass spectrometry or Western blotting. A unique feature of PB is its ability to capture even transient or low-affinity interactions, making it especially valuable for studying interactions between protein co-regulators and transcription factors [6]. For example, the BioID ligase has been used to identify specific partners of the transcription factor Ubx in various embryonic tissues of *Drosophila* [7]. Our research group has also previously conducted a study aimed at identifying novel protein partners of the ecdysone receptor (EcR) in *Drosophila* S2 cells of embryonic origin using BioID2 and APEX2 techniques. Among the detected proteins were both previously known partners—such as nuclear pore components Nup358, Nup214, Nup88, Nup50, and Nup205—and proteins whose involvement in the ecdysone response had not been described before, such as the architectural protein CP190 [8].

Many PB methods are time-consuming, often requiring more than 18 h for effective biotinylation, or can be toxic to tissues [9]. As a result, most PB approaches are unsuitable for studying short-term protein–protein interactions, which are characteristic of transcription factors. The use of TurboID—the most active and sensitive biotin ligase currently available—addresses these limitations, as it enables rapid and efficient biotinylation [9].

Streptavidin-based (Str) purification is a classic and highly effective method for enriching biotinylated proteins. This technique takes advantage of one of the strongest non-covalent interactions in nature, enabling the use of stringent washing conditions during purification. However, Str purification has a limitation: ‘on-beads’ digestion of enriched proteins often results in contamination by streptavidin-derived peptides. This issue can be addressed by performing ‘in-solution’ digestion prior to biotin enrichment [10,11].

In this study, we employed a proximity biotinylation (PB) method utilizing the promiscuous biotin ligase TurboID to identify protein partners of the *Drosophila melanogaster* ecdysone receptor (EcR) in ovarian cells. EcR is a key transcription factor essential for *Drosophila* development and metamorphosis [12]. It enables various tissues to respond differentially to increases in 20-hydroxyecdysone (20E) concentrations by activating distinct sets of genes and interacting with diverse tissue-specific coregulators [13]. Because interactions between nuclear receptors and their coregulators are typically highly specific but relatively weak, many EcR coregulators likely remain unidentified—a gap that PB techniques are well-suited to address [14].

The plastic response of *Drosophila* tissues to 20E highlights the need for comparative identification of EcR coregulatory partners across different tissues, often working with limited amounts of material. To facilitate the identification of ecdysone receptor partners in various *Drosophila* tissues, we optimized a protocol for biotinylated protein purification and evaluated different strategies for enriching biotinylated protein fractions.

## 2. Results

### 2.1. Direct Comparison of Methods for Precipitation and Elution of Biotinylated Proteins

To determine the most effective approach for enriching small amounts of biotinylated proteins from *Drosophila* tissues, we first conducted a series of experiments to purify endogenous biotinylated proteins from *Drosophila* Schneider 2 (S2) cells under different conditions (Figure 1A). Specifically, we evaluated two precipitation methods: (1) streptavidin–agarose (“Str”) and (2) anti-biotin antibodies immobilized on Protein A-sepharose (“Biotin Abs”), with antibodies either non-covalently or covalently bound using DMP (dimethyl pimelimidate) cross-linking.

Both methods successfully precipitated biotinylated proteins. However, “Str” (streptavidin–agarose) demonstrated greater precipitation efficiency compared to “Biotin Abs” immobilized either non-covalently (DMP−) or covalently (DMP+) on Protein A-sepharose. The reduced efficiency observed with covalently immobilized antibodies (DMP+) compared to (DMP−) is likely due to partial inactivation of anti-biotin antibodies during DMP cross-linking. Although precipitation using “Biotin Abs” was less efficient than “Str,” the DMP-based antibody immobilization effectively removed antibodies from the eluted sample in the (DMP+) condition.

Furthermore, “Str”-based precipitation completely depleted biotinylated proteins from the extract, unlike the antibody-based approach. The main drawback of “Str”-based precipitation was the presence of significant amounts of streptavidin in the eluted sample, as revealed by Coomassie staining (Figure 1B).

After detecting substantial amounts of streptavidin in the eluted samples, we hypothesized that lowering the elution temperature might help to strike a better balance between elution efficiency and sample contamination with streptavidin. To test this, we evaluated three different elution temperatures (Figure 2A). As expected, the yield of biotinylated proteins decreased with lower elution temperatures for both “Str” and “Biotin Abs” (DMP+) samples. Simultaneously, the amount of streptavidin in the eluates also decreased as the temperature was lowered, nearly disappearing at 50 °C (Figure 2B). Based on these results, 75 °C appears to offer an optimal balance: approximately 50% of the biotinylated proteins were recovered, while streptavidin contamination in the eluate was reduced by more than half. Furthermore, elution at 75 °C was also effective for recovering biotinylated proteins from “Biotin Abs” (DMP+) preparations, as the protein yield was comparable to that obtained at 95 °C.

The aim of our study was to develop an effective method for precipitating and detecting small amounts of biotinylated proteins from *Drosophila* tissues using LC-MS. Since trypsinolysis can be employed to elute precipitated proteins, streptavidin contamination in the eluate can be avoided by using a trypsin-resistant streptavidin resin. To achieve this, we adapted a previously described protocol for the chemical modification of streptavidin, which renders it resistant to cleavage by trypsin and LysC proteases [15]. This modification prevents contamination of samples with short streptavidin-derived peptides without compromising the binding affinity of streptavidin.

We modified streptavidin–agarose according to the published protocol and compared the effectiveness of regular-type streptavidin (rtS) and protease-resistant streptavidin (prS) in precipitating biotinylated proteins from *Drosophila* S2 cell lysates. Precipitation experiments with endogenous biotinylated proteins from S2 cells showed that rtS and prS performed comparably (Figure 3). Both types of sepharose successfully depleted most biotinylated proteins from the cell lysate. The precipitated proteins were directly digested with trypsin on the sepharose. LC-MS/MS analysis revealed that streptavidin-derived peptides constituted 87.9% of the proteome in the rtS sample, compared to only 14.7% in the prS sample (as determined using PEAKS® Studio 11 software, for Streptavidin ID: P22629). Thus, prS achieved the same efficiency of protein depletion as rtS while significantly reducing streptavidin peptide contamination in the trypsin-eluted sample. Based on these results, we selected chemically modified streptavidin–sepharose (prS) as the optimal method for enriching biotinylated proteins from *Drosophila* protein extracts.

### 2.2. Identification of EcR Protein Partners in Drosophila Tissues Using the TurboID Biotinylating Technique

The EcR gene encodes three isoforms—EcR-A, EcR-B1, and EcR-B2—which possess identical DNA-binding and ligand-binding domains but differ in their N-terminal regions [16]. RNA-Seq data indicate that *Drosophila* ovarian tissue does not express the EcR-B1 and EcR-B2 isoforms, as no RNA-Seq signal is detected at the respective exons (Figure 4). The predominant EcR isoform present in the ovaries is EcR-A, which was, therefore, selected as the bait for our PB experiments. In contrast, both EcR-B1 and EcR-B2 isoforms are abundantly expressed in salivary glands. These results demonstrate that EcR isoform expression is tissue-specific in *Drosophila*.

To perform proximity biotinylation of EcR protein partners in *D. melanogaster* tissues, we raised a transgenic *Drosophila* stock expressing the EcR-A isoform fused to the TurboID biotin ligase (TurboID-EcR stock) (Figure 5). As a negative control, we used a transgenic stock carrying a similar construct but lacking the EcR-A coding region (TurboID-Gly stock) [8,9]. Detailed descriptions of the plasmids used to generate these transgenic flies are provided in the Materials and Methods section.

For each transgenic stock, transgene expression was driven in ovarian somatic cells. Lysates were prepared from dissected ovaries in two independent biological replicates and subjected to protein enrichment. Prior to enrichment, the lysates were verified for expression of the transgenic proteins and successful biotinylation (Figure 6). In both samples, the protease-resistant streptavidin (prS) resin completely depleted biotinylated proteins from the extracts. As TurboID ligase was expected to biotinylate the expressed transgenic protein, we checked for its presence among the precipitated proteins. As anticipated, the precipitated TurboGly sample contained a high amount of FLAG-tagged protein with an apparent molecular weight of 35–40kDa, consistent with the predicted size of TurboID. The TurboEcR fusion protein migrated according to its expected calculated size as well.

Given that test precipitation from ovarian *Drosophila* extracts confirmed the expression, biotinylation, and successful precipitation of the transgenic TurboEcR and TurboGly proteins using prS, we subjected the remaining lysate to prS precipitation, followed by trypsinolysis and LC-MS/MS analysis. Each sample was analyzed in two independent biological replicates, enabling quantitative statistical assessment of protein presence. The results revealed specific enrichment of proteins in the TurboEcR sample compared to the TurboGly control (Figure 7). Among the statistically significantly enriched proteins—putative EcR-A partners in ovarian tissue—we identified several nuclear and DNA-binding proteins, highlighted in orange and red, respectively (Appendix A).

## 3. Discussion

*Drosophila melanogaster* serves as a valuable experimental model for testing candidate genes identified in human screens to assess their functional contributions to pathological conditions [1,18]. The availability of numerous mutant stocks enables comprehensive genetic studies of gene interactions in *Drosophila*. Its advanced genetic manipulation systems also make *Drosophila* a promising model for probing protein–protein interactions using proximity biotinylation (PB). Transgenic lines expressing a protein of interest fused to a biotinylating enzyme for PB can be generated rapidly, and the extensive database of driver stocks allows for targeted expression in virtually any tissue—even in individual cells. However, a significant limitation of using *Drosophila* for studying protein–protein interactions is the necessity to work with a limited amount of biological material.

### 3.1. Pros and Cons of Different Approaches to Enrich Biotinylated Proteins from Drosophila Tissues (With Limited Material Availability)

At various stages of proximity biotinylation (PB) methodology for studying protein–protein interactions in *Drosophila*, different enrichment strategies can be employed, each with its own advantages and limitations.

#### 3.1.1. Protein Extraction

A rich panel of promiscuous biotinylating enzymes now enables efficient *in vivo* labeling of partner proteins in *Drosophila* tissues [19]. Earlier enzymes were not suitable for this purpose, as they required higher operating temperatures than *Drosophila* cells can tolerate. However, selecting an optimal approach for the efficient extraction of biotinylated proteins from tissue remains a significant challenge. For many years, RIPA buffers containing both ionic and non-ionic detergents have been considered the most effective means for protein extraction [9]. These buffers facilitate the extraction of both cytoplasmic and nuclear proteins while disrupting protein complexes. This property is advantageous for PB experiments, as it ensures that only proteins in close proximity to the bait—those directly labeled with biotin—are recovered, rather than indirect interactors.

Efficient protein extraction is especially critical when working with limited material; insufficient extraction can lead to underrepresentation of proteins and incomplete partner identification. Unfortunately, the detergents present in RIPA buffers interfere with the quality of protein detection by LC-MS/MS. It is therefore necessary to thoroughly remove detergents both during the precipitation of biotinylated proteins and after the peptide loading on a reversed-phase column for MS/MS analysis. Whenever possible—such as when extracting predominantly cytoplasmic proteins—the use of buffers with lower detergent concentrations is preferable.

#### 3.1.2. Protein Enrichment and Elution

The most effective method for enriching biotinylated proteins from extracts is the use of streptavidin-loaded resins. The exceptionally high affinity of the biotin–streptavidin interaction allows for the application of stringent washing conditions and results in a high degree of enrichment. However, eluting biotinylated proteins from immobilized streptavidin is extremely challenging unless biotinylation is performed with iminobiotin, which can be displaced under milder conditions [20]. Unfortunately, iminobiotin cannot be used for proximity biotinylation (PB) since biotin ligases do not efficiently incorporate it *in vivo* [21]. Efficient elution of proteins from resins with non-covalently bound streptavidin, whether by heating or direct trypsinolysis, typically results in sample contamination with streptavidin-derived peptides. This is particularly problematic with low-input material, as the streptavidin signal can obscure the detection of target proteins that are present in minor amounts during MS/MS analysis. When detection by mass spectrometry is the primary aim, this issue can be addressed by chemically or genetically modifying streptavidin to make it resistant to proteases such as trypsin, thus reducing streptavidin peptide contamination in the analyzed sample [15,22]. Our findings confirm previous reports that protease-resistant streptavidin maintains strong affinity for biotin while substantially reducing streptavidin-derived contamination. A notable drawback of this approach, however, is the absence of directly biotinylated peptides in the eluate, resulting in a loss of information regarding the specific biotinylation sites on target proteins, since the biotin–streptavidin interaction remains intact under these elution conditions.

Another widely used approach for enriching biotinylated proteins is precipitation with anti-biotin antibodies. However, our data demonstrate that antibody-based precipitation does not achieve the same efficiency as streptavidin-based enrichment. This limitation is especially problematic when working with small amounts of material, where maximal recovery is critical. In situations where proteins biotinylated by the exogenous enzyme represent only a small fraction of the total endogenous biotinylated proteins, highly efficient depletion of the extract is essential for the comprehensive detection of all target proteins.

#### 3.1.3. Trypsinolysis

There are two main approaches to the trypsinolysis of biotinylated proteins, each yielding fundamentally different results: digestion in solution before enrichment, and digestion on beads after enrichment. Recently, in-solution digestion prior to enrichment has gained popularity, as it can increase the complexity of the resulting peptide library [23,24]. However, when applying this approach to proximity biotinylation (PB) methods, its effectiveness depends on a high proportion of biotinylated peptides derived from the target protein; otherwise, there is a risk of failing to identify certain targets. Since limited starting material already poses a challenge for adequate peptide coverage and target identification, the in-solution digestion approach is generally unsuitable in such cases.

#### 3.1.4. General Observations

Biotinylation of proteins in tissues using PB methods can significantly impact the physiology and viability of *Drosophila*. In our experiments focused on protein biotinylation in ovaries, we did not observe a significant reduction in fly vitality—likely due to the supplementation of excess biotin in the diet. However, in some earlier experiments, we noted a reduction in the levels of endogenously biotinylated proteins in the control TurboGly flies when transgene expression was higher, which we attribute to the onset of biotin deficiency. We propose that the presence of endogenously biotinylated proteins serves as a useful marker of physiological biotinylation in tissues. Additionally, we observed that the control, shorter TurboGly protein was expressed at a higher level, resulting in increased non-specific biotinylation background. Based on these findings, we suggest that using an inert protein—such as GFP or dCas9—fused to a biotin ligase as a control line for assessing background biotinylation is a more appropriate strategy [25]. This approach may help enhance the identification rate of genuine targets.

### 3.2. PB Helped to Identify Previously Unknown Protein Partners of EcR in Drosophila Ovary

We investigated whether the use of protease-resistant streptavidin, in combination with the TurboID PB technique, would help to identify EcR protein partners in *Drosophila* ovarian tissue. EcR is an especially promising target for proximity labeling studies in *Drosophila*, as different tissues display distinct responses to ecdysone, and the molecular mechanisms underlying these variations remain largely unexplored. Using TurboID labeling, we successfully identified both previously described and novel EcR-associated partners in *Drosophila* ovaries (Figure 8). Notably, among the proteins biotinylated with EcR as bait, we detected two corepressors—Smr and CtBP—that have previously been implicated in EcR-mediated repression [8,17]. We propose that these proteins exert their repressive roles by cooperating with EcR, particularly when 20E levels are low.

The identified EcR partners included several subunits of the transcriptional machinery: Rpb8 of RNA polymerase II; Mbf1, a coactivator associated with TBP; dSki8 of the PAF complex involved in transcriptional elongation; CstF, a component of the polyadenylation machinery; and tws, a subunit of the INTAC complex that participates in the release of transcriptional “pausing” [26,27,28,29,30,31]. The purification of these components alongside EcR demonstrates that the activation of 20E-dependent transcription is closely linked to direct interactions between the nuclear receptor complex and various elements of the transcriptional apparatus. This finding supports our recent hypothesis that the positive effect of 20E on transcription is mediated, at least in part, through direct interactions between EcR and components of the pre-initiation complex (PIC), thereby promoting the recruitment of TBP and even RNA polymerase II itself [13].

Some of the newly identified EcR partner proteins may help elucidate aspects of *Drosophila* models for human pathologies. For instance, the binucleate secondary cells (SCs) of the *Drosophila* male accessory gland are considered a model for human castrate-resistant prostate cancer. The E2F1 transcription factor plays a pivotal role in promoting the proliferation of both SCs and prostate cancer cells by regulating cell cycle gene expression [32]. In humans, E2F1 collaborates with the androgen receptor (AR) to modulate transcription in a hormone-independent manner, whereas in *Drosophila*, E2F1 functions together with EcR irrespective of 20E [32]. In our study, we demonstrated a direct protein interaction between EcR and Dp (the dimerization partner of E2F1), indicating that EcR and E2F1 can interact not only functionally but also physically at enhancer or promoter regions. Additionally, we identified a previously unreported interaction between EcR and Peb, the *Drosophila* ortholog of the human RREB transcription factor, which is involved in androgen-dependent carcinogenesis. This finding may further advance the utility of *Drosophila* as a model for studying hormonally regulated cancer phenotypes [33].

Our identification of EcR protein partners adds to the existing knowledge of ecdysone receptor-associated proteins. It is important to note that the PB method labels not only proteins that interact directly with the bait, but also those in close proximity; thus, we do not assume that all identified proteins are direct interactors. Instead, we propose that EcR and its partners are components of common multiprotein complexes within the cell. Further studies are necessary to functionally validate these interactions and clarify their specific contributions to *Drosophila* physiology.

## 4. Materials and Methods

**Biotin antibodies for the enrichment of biotinylated proteins.** Biotin antibodies (Abcam ab53494) were covalently immobilized on Mab-select sepharose (GE Healthcare) and used for affinity purification of biotinylated proteins. The immobilization procedure included antibody incubation, cross-linking with dimethyl pimelimidate (DMP), blocking with glycine, and washing in RIPA buffer (see details in Appendix A). RIPA buffer was selected for these washes due to its effectiveness in nuclear protein extraction [9]. An experimental workflow is schematically presented in Figure 9.

**Cell and tissue lysate precipitation.** Lysates from *D. melanogaster* S2 cells and ovaries were prepared by homogenization in RIPA buffer followed by sonication and DNase I treatment. Ovarian tissues were manually dissected, frozen at −70 °C, and processed similarly after thawing. Lysates were cleared by centrifugation and used for precipitation with either sepharose with biotin antibodies attached or streptavidin–agarose (Sigma-Aldrich). After precipitation, sepharose was washed 3 times with RIPA buffer. Precipitated proteins were eluted with SDS 2% elution buffer (see details in Appendix A).

**Protease-resistant streptavidin sepharose (prS).** To produce protease-resistant streptavidin, the previously described protocol was used [15]. PrS was produced by chemical modification of streptavidin–agarose (Sigma-Aldrich, Burlington, Massachusetts, United States) using 1,2-cyclohexanedione and formaldehyde with sodium cyanoborohydride. This treatment improves resistance to proteolysis during on-bead digestion (see details in Appendix A).

**Transgenic *Drosophila melanogaster***. The coding region of EcR-A was fused with TurboID ligase (TurboEcR) tagged with 3×FLAG and inserted into the *attP2* site under control of a 10×UAS-hsp43 promoter [34,35,36]. The negative control line (TurboGly) lacked the EcR-A coding region but carried the same vector backbone. Expression in ovarian somatic cells was driven by *tj-GAL4* (AA274 stock previously obtained by A. Aravin). Females were raised on biotin-supplemented food for 24 h prior to ovary dissection. For each genotype, 120 ovaries were collected (60 per biological replicate), processed for protein extraction and enrichment, and validated by Western blot analysis. The overall experimental design for protein enrichment from *Drosophila* ovarian tissue is summarized in Figure 10.

**Peptide fingerprinting and LC-MS/MS analysis.** To quantitatively compare proteins precipitated from TurboEcR versus TurboGly tissues, lysates were prepared in parallel, enriched using prS, washed with PBS, and subjected to on-bead trypsinolysis. Reduction, alkylation, and digestion were performed using sodium deoxycholate-based buffer, followed by peptide purification on SDB-RPS StageTips and elution with acetonitrile/ammonia [37]. Peptides were vacuum-dried, stored at −80 °C, and reconstituted in 2% acetonitrile/0.1% TFA prior to LC-MS/MS analysis. Data were processed with MaxQuant (https://www.maxquant.org, version 2.4.2.0), and downstream analysis (including differential quantification, GO enrichment, and Volcano plot generation) was performed using custom Python (3.11.5) scripts (Pandas (2.0.3), NumPy (1.24.3), Matplotlib (3.7.2), Plotly (5.9.0), adjustText (1.2.0) (see details in Appendix A).

## Figures and Tables

**Figure 1 ijms-26-08009-f001:**
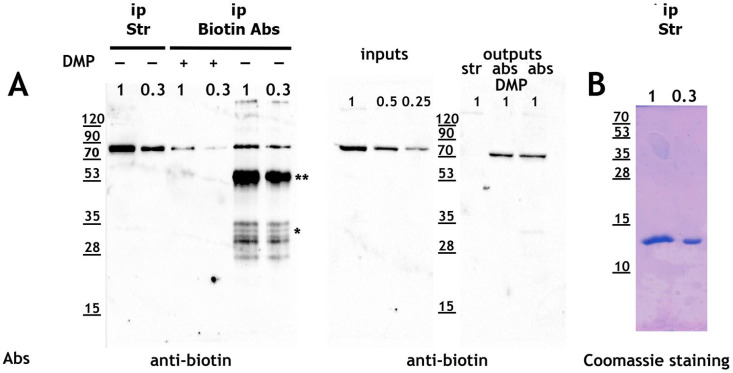
**Biotin antibodies are less efficient than streptavidin–agarose for precipitating biotinylated proteins.** (**A**,**B**): Protein lysates from S2 cells were subjected to precipitation using streptavidin–agarose (ip Str) and (Mab)-sepharose complexes with biotin antibodies (ip Biotin Abs). Both the initial protein lysate (inputs) and the precipitated fractions (ip) were analyzed in dilution, while the unprecipitated fractions (outputs) were analyzed undiluted. Biotinylated proteins were eluted by incubating the resins in 2% SDS elution buffer at 95 °C for 5 min. Western blotting was performed using anti-biotin antibodies, and precipitates from streptavidin–agarose were also visualized by Coomassie staining (panel **B**). Abbreviations: *—light chains of antibodies, **—heavy chains of antibodies, inputs—source protein lysate, outputs—unprecipitated lysate, ip—precipitate.

**Figure 2 ijms-26-08009-f002:**
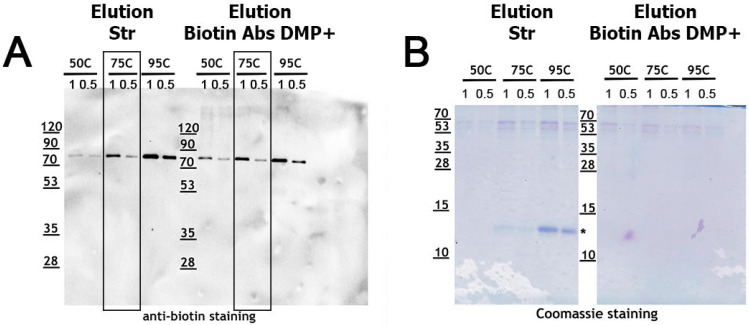
**Increasing elution temperature enhances the recovery of biotinylated proteins and streptavidin monomers in the eluate.** (**A**): Elution profiles following precipitation of protein lysate from *Drosophila* S2 cells using streptavidin–agarose (Elution Str) and Mab-sepharose with biotin antibodies (Elution Biotin Abs DMP+) at various temperatures. Eluates were serially diluted and analyzed by Western blot using anti-biotin antibodies. (**B**): The same gel was stained with Coomassie Blue to visualize streptavidin monomer. Designation: *—streptavidin monomer.

**Figure 3 ijms-26-08009-f003:**
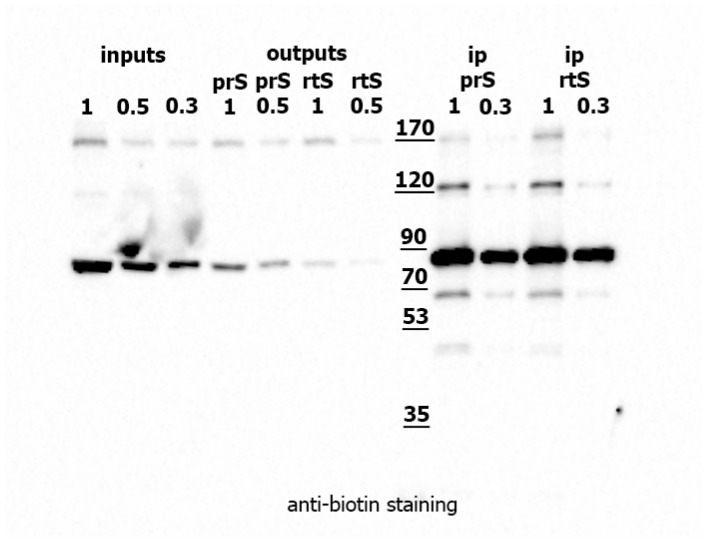
**Protease-resistant streptavidin (prS) depletes biotinylated proteins from *Drosophila* S2 cell lysate as efficiently as regular streptavidin (rtS).** Protein lysates from *D. melanogaster* S2 cells were subjected to precipitation using either protease-resistant streptavidin–sepharose (prS) or regular streptavidin–agarose (rtS). The original lysate (inputs), unprecipitated fractions (outputs), and precipitated fractions (ip) were loaded in dilution. Western blotting was performed with anti-biotin antibodies to detect biotinylated proteins.

**Figure 4 ijms-26-08009-f004:**
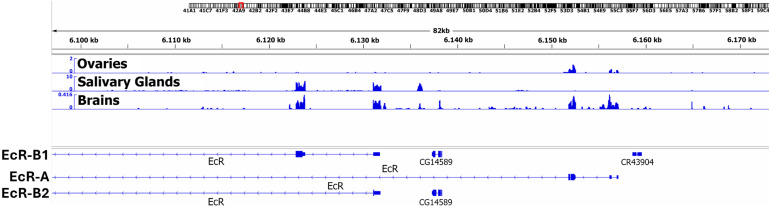
**Expression of the ecdysone receptor isoforms (EcR-A/EcR-B1/EcR-B2) in various tissues of *D.melanogaster* (ovaries/salivary glands/brains).** RNA-seq analysis was performed on the polyA-enriched RNA fraction from salivary glands and brains of late third-instar wandering *Drosophila* larvae [13]. RNA-seq analysis of ovaries (stages 1–16 of oogenesis) was published previously by our group [17]. The figure presents RNA-seq data visualized in the IGV genome browser, with signal shown as normalized read counts per genome size. Transcript schematics for each EcR isoform are displayed below the RNA-seq tracks.

**Figure 5 ijms-26-08009-f005:**
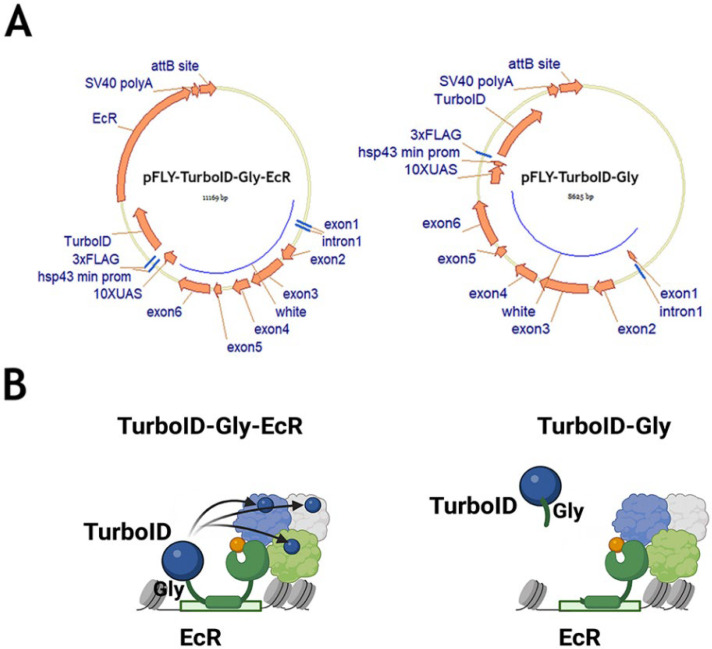
**Plasmids for expression of target (EcR) and TurboID-fused proteins in *D. melanogaster* tissues.** (**A**): Schematic diagrams of plasmids used in this study. The TurboEcR plasmid (pFLY-TurboID-Gly-EcR) encodes the ecdysone receptor (EcR) fused to TurboID biotin ligase, while the TurboGly plasmid (pFLY-TurboID-Gly) expresses TurboID with a Gly linker and lacks the EcR coding region, serving as a negative control. The locations of the triple FLAG epitope tag (3x FLAG), 10xUAS, and the minimal *hsp43* promoter (hsp43 min prom) are indicated. (**B**): Model illustrating the mechanism of proximity biotinylation mediated by TurboEcR (TurboID-Gly-EcR) and TurboGly (TurboID-Gly negative control) fusion proteins (created with BioRender.com).

**Figure 6 ijms-26-08009-f006:**
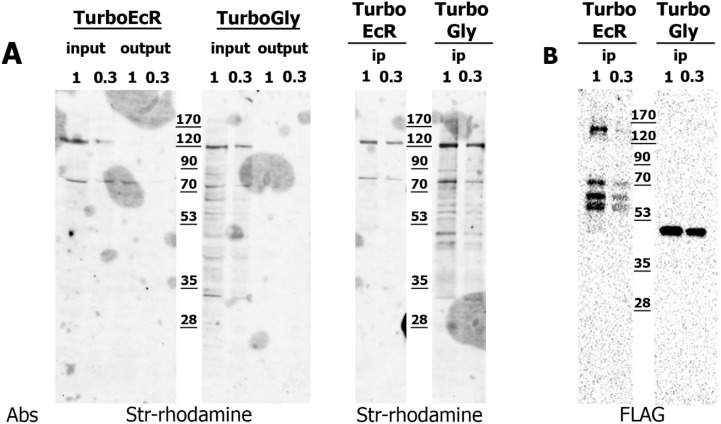
**Expression and biotinylation activity of TurboEcR and TurboGly fusion proteins in ovarian tissues of transgenic *D. melanogaster* strains.** (**A**): Lysates from ovarian tissues of flies expressing TurboEcR (TurboEcR input) or TurboGly (TurboGly input) were subjected to protein precipitation using protease-resistant streptavidin (prS). The unprecipitated fractions (TurboEcR output, TurboGly output) showed complete depletion of biotinylated proteins. Precipitated biotinylated proteins from both the TurboEcR (TurboEcR ip) and TurboGly (TurboGly ip) strains were analyzed by Western blot using streptavidin-rhodamine conjugate (Abs Str-rhodamine). (**B**): Presence of transgenic proteins in the precipitated fractions (TurboEcR ip and TurboGly ip) was confirmed by Western blot using anti-FLAG antibodies (Abs FLAG).

**Figure 7 ijms-26-08009-f007:**
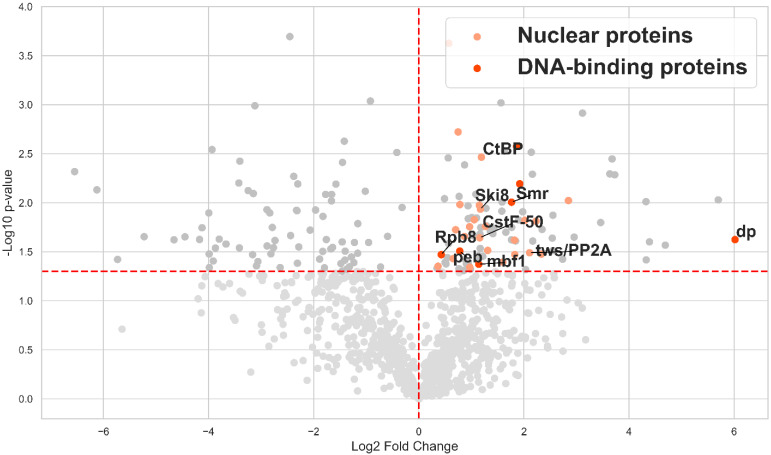
**Identification of EcR-A partners in *Drosophila* ovaries using proximity biotinylation.** Volcano plot based on LC-MS/MS analysis of biotinylated proteins purified via EcR-TurboID chimeric protein in ovarian tissue of *D. melanogaster*. Gene names correspond to proteins detected by LC-MS/MS. The *X*-axis represents Log2 Fold Change (enrichment relative to TurboGly control; calculated as the log2 fold change between TurboEcR and TurboGly samples). Positive Log2 Fold Change values indicate proteins specifically enriched in TurboEcR samples. The *Y*-axis represents the negative log10 of the *p*-value (t-criteria) for each protein. The vertical red line marks Log2 Fold Change = 0, while the horizontal red line indicates the significance threshold (*p*-value ≤ 0.05).

**Figure 8 ijms-26-08009-f008:**
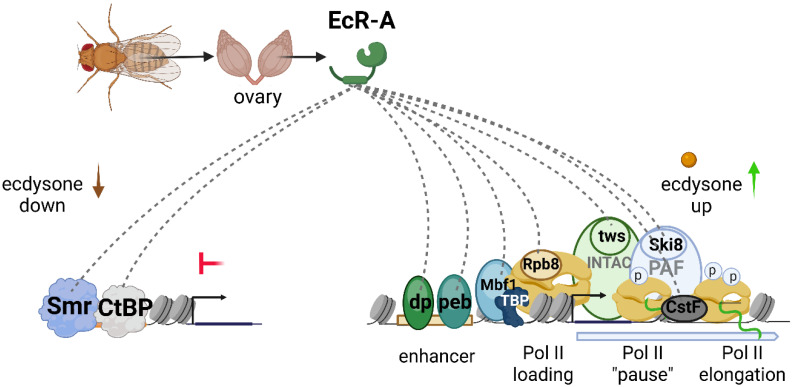
**Model of EcR protein complex in *Drosophila’s* ovaries.** EcR protein partners (identified in the current study), involved in transcription activation and repression, are shown (image created with BioRender.com).

**Figure 9 ijms-26-08009-f009:**
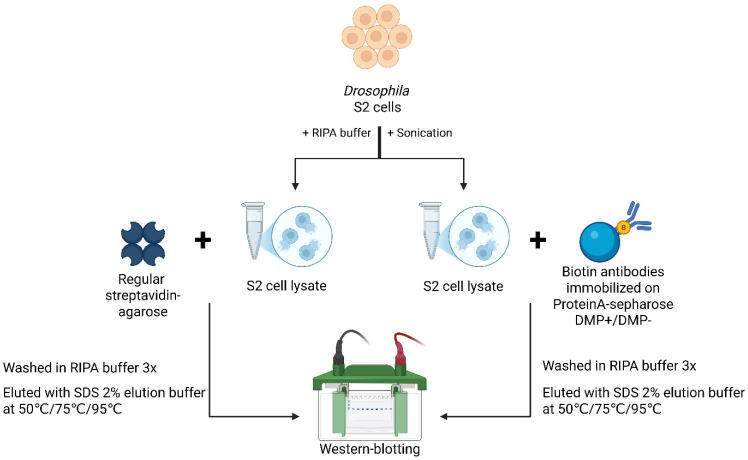
Experimental procedures performed for the enrichment of biotinylated proteins from *Drosophila* S2 cell lysates using streptavidin–agarose or antibody-based affinity purification. S2 cells were lysed in RIPA buffer, and biotinylated proteins were precipitated using either regular streptavidin–agarose or biotin antibodies immobilized on Protein A-sepharose via DMP-mediated cross-linking. After washing, bound proteins were eluted at increasing temperatures and analyzed by Western blot (image created with BioRender.com).

**Figure 10 ijms-26-08009-f010:**
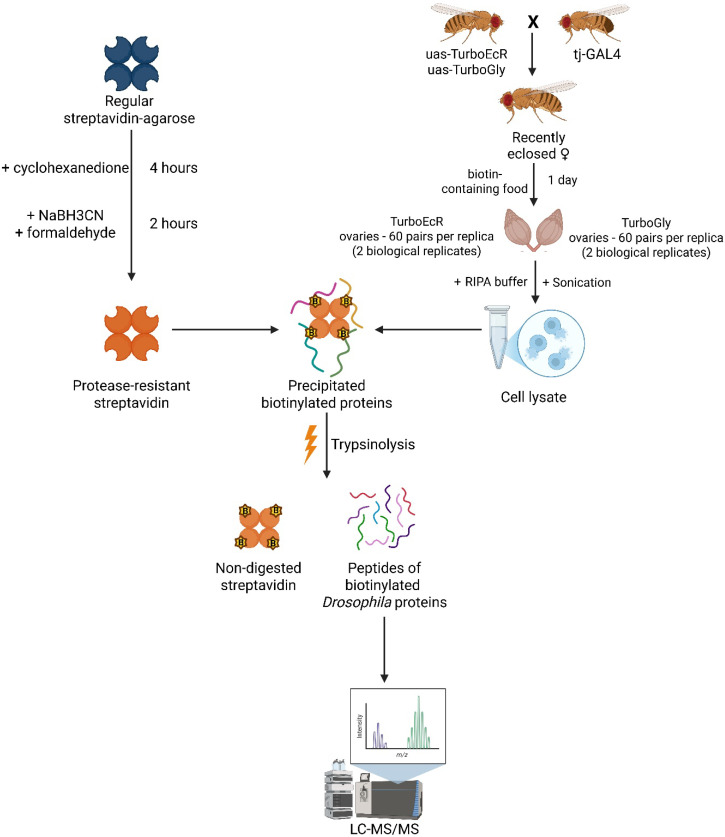
**Workflow of the TurboID-based proteomic profiling in *Drosophila* ovaries.** Transgenic females expressing TurboID-tagged EcR-A (TurboEcR) or control construct (TurboGly) in ovarian somatic cells were raised on biotin-containing food. Ovaries were dissected, lysed, and biotinylated proteins were enriched using protease-resistant streptavidin (prS). After tryptic digestion, peptides were analyzed by LC-MS/MS (image created with BioRender.com).

## Data Availability

The original contributions presented in this study are included in the article/Appendix A. Further inquiries can be directed to the corresponding author.

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
