# Peer review of "Adaptation of the Protocol for the Isolation of Biotinylated Protein Complexes for Drosophila melanogaster Tissues"

_ijms, 2025, doi:10.3390/ijms26168009_

Round 1

Reviewer 1 Report

Comments and Suggestions for Authors

Proximity biotinylation using enzymes like BioID and TurboID is a powerful method for identifying protein-protein interactions but is limited by high background noise from streptavidin-based enrichment, especially in small tissue samples. This study compared methods for purifying biotinylated proteins from Drosophila melanogaster S2 cells and tissues, finding that protease-resistant streptavidin (prS) offers similar efficiency with reduced background noise compared to regular streptavidin. Using prS and TurboID, the researchers successfully identified known and novel protein partners of the ecdysone receptor in Drosophila ovarian tissue, demonstrating the method’s effectiveness. The study will be of interest to researchers using proximity labeling in their studies. Below are some technical comments that the authors may consider in their revisions.

It is somewhat unfortunate that RIPA buffer was used for washing and lysis. One of the detergents in RIPA, NP-40, is difficult to completely remove and may have reduced the sensitivity in mass spectrometry. It would have been valuable to assess whether the presence of such detergents affected the enrichment process or the mass analysis, and to evaluate their impact systematically.

The issue of streptavidin contamination by trypsin typically arises when on-bead digestion is performed. If in-solution digestion is carried out prior to streptavidin enrichment, there is no concern about streptavidin degradation by proteases. At the very least, it would have been helpful to reference studies that avoid this issue by performing enrichment at the peptide level (e.g., PMID: 28156110, 29039416, 38724559). That said, protein-level enrichment has its own advantages, and in this context, the use of protease-resistant streptavidin represents a meaningful improvement.

Although the authors report an 85-fold reduction in streptavidin-derived peptides in the prS versus rtS comparison, it would have been more informative to also present quantitative metrics such as total peptide IDs, enrichment fold changes, and proteome coverage.

Rather than simply identifying all proteins present in the IP eluates, it would be more accurate to specifically examine proteins with lysine residues that are biotinylated. This would better reflect the true proximity labeling targets.

With only two biological replicates, it is difficult to place statistical confidence in the comparative analysis. At least three replicates are generally required to support reliable statistical comparisons of protein identification and mass intensity values.

The candidate proteins identified by proximity labeling only indicate spatial proximity to the bait protein. This does not necessarily imply direct interaction or functional association with EcR. To support these findings, additional validation would be helpful—such as the other interaction prediction tools, co-immunoprecipitation assays, or imaging-based evidence of co-localization.

Author Response

Proximity biotinylation using enzymes like BioID and TurboID is a powerful method for identifying protein-protein interactions but is limited by high background noise from streptavidin-based enrichment, especially in small tissue samples. This study compared methods for purifying biotinylated proteins from Drosophila melanogaster S2 cells and tissues, finding that protease-resistant streptavidin (prS) offers similar efficiency with reduced background noise compared to regular streptavidin. Using prS and TurboID, the researchers successfully identified known and novel protein partners of the ecdysone receptor in Drosophila ovarian tissue, demonstrating the method’s effectiveness. The study will be of interest to researchers using proximity labeling in their studies. Below are some technical comments that the authors may consider in their revisions.

We are very grateful to the Review for finding our research interesting and for the valuable comments, with which we fully agree. 

It is somewhat unfortunate that RIPA buffer was used for washing and lysis. One of the detergents in RIPA, NP-40, is difficult to completely remove and may have reduced the sensitivity in mass spectrometry. It would have been valuable to assess whether the presence of such detergents affected the enrichment process or the mass analysis, and to evaluate their impact systematically.

Indeed, we used RIPA buffer (50 mM Tris; 150 mM NaCl; 0.1% SDS; 0.5% sodium deoxycholate; 1% Triton X-100; pH 7.5), which lacks NP-40 but contains a number of different ionic and non-ionic detergents, to extract proteins from Drosophila cells and tissues. We fully acknowledge that the use of detergents may interfere with subsequent LC-MS/MS detection. We believe, however, that in the case of studies such as ours, where proteomes of DNA-binding proteins (like EcR) are studied and where it is necessary to extract chromatin-associated proteins from tissues, the use of harsh extraction conditions is practically impossible to avoid. For our part, we attempted to wash out detergents in the downstream parts of the protocol: (1) before ‘on-bead’ digestion of the precipitated biotinylated proteins, after three washes with RIPA, we additionally washed the streptavidin beads with PBS three times; (2) after loading the peptides onto the reversed-phase column before MS/MS, we additionally washed the column with ethyl acetate, which should also remove traces of detergents.

The issue of streptavidin contamination by trypsin typically arises when on-bead digestion is performed. If in-solution digestion is carried out prior to streptavidin enrichment, there is no concern about streptavidin degradation by proteases. At the very least, it would have been helpful to reference studies that avoid this issue by performing enrichment at the peptide level (e.g., PMID: 28156110, 29039416, 38724559). That said, protein-level enrichment has its own advantages, and in this context, the use of protease-resistant streptavidin represents a meaningful improvement.

We are really grateful to the Reviewer for this valuable suggestion. Indeed, we were not familiar with the ‘in solution’ trypsinolysis prior to precipitation. Although we do not suppose that this is the method of choice for precipitating small amounts of biotinylated proteins from Drosophila tissues after proximal biotinylation (PB). For protein to be identified after fingerprinting is necessary to have some, better several peptides in the sample. In the case of in-solution processing after PB, this means the need for efficient in vivo protein biotinylation (each protein into at least several peptides) and precipitation of these peptides in quantities sufficient for detection. The latter is difficult to achieve when the amount of material is limited (in the case of Drosophila tissues). We believe that it is precisely in our case, when there is insufficient material, that the presence of additional non-biotinylated peptides can facilitate the identification of additional hits. Of course, we believe that this assumption is best tested experimentally. However, we do not think that there is room for this in the current manuscript. Although we discussed the possibility of ‘in solution’ digestion approach in the new version of Discussion (lines 301-309)

 Although the authors report an 85-fold reduction in streptavidin-derived peptides in the prS versus rtS comparison, it would have been more informative to also present quantitative metrics such as total peptide IDs, enrichment fold changes, and proteome coverage.

In the first version of the manuscript, we evaluated the presence of streptavidin in the samples by comparing the area values (which gave us 85-fold reduction). In preparing the revised version, we performed a more accurate analysis and determined the proportion of the proteome represented by streptavidin in the sample (analysis was performed using Peaks software). The result was not so impressive however we detected a strong decrease in the content of streptavidin in the prS sample. The proportion of streptavidin in the proteome identified for the rtS sample was 87.9%, while for the prS sample it was 14.7% (estimated using Peaks software for Streptavidin ID:P22629). This last characteristic was added to the manuscript (lines 149-152).

Rather than simply identifying all proteins present in the IP eluates, it would be more accurate to specifically examine proteins with lysine residues that are biotinylated. This would better reflect the true proximity labeling targets.

We would certainly use this valuable advice if we were eluting proteins from a resin carrying non-covalently linked streptavidin using detergents. Unfortunately, our ‘on-beads’ digestion protocol uses elution conditions under which the interaction between biotin and streptavidin cannot be broken. This is why biotinylated peptides remain bound to streptavidin and are absent from the sample subjected to MS analysis. We checked for the presence of biotinylated peptides in the sample analyzed by MS. However, as expected, they were absent. 

With only two biological replicates, it is difficult to place statistical confidence in the comparative analysis. At least three replicates are generally required to support reliable statistical comparisons of protein identification and mass intensity values.

We fully agree with the Reviewer. Of course, it would be much better to have at least three biological replicates for each sample. Unfortunately, financial constraints and the limited amount of material for analysis did not allow us to do so. We would like to note that in our work we use proteomic analysis by PB in order to find new potentially interesting partners suitable for further biochemical and functional analysis (including the obtaining of specific antibodies). Thus, we do not make deep conclusions about the structural and functional relationship between proteins based only on proteomic analysis. We try to supplement our analysis with other experiments as much as possible. We position this manuscript as a presentation of a methodological approach that allows the analysis of protein interactions in Drosophila tissues.

 The candidate proteins identified by proximity labeling only indicate spatial proximity to the bait protein. This does not necessarily imply direct interaction or functional association with EcR. To support these findings, additional validation would be helpful—such as the other interaction prediction tools, co-immunoprecipitation assays, or imaging-based evidence of co-localization.

We fully agree with the Reviewer regarding this statement. We added a sentence mentioning that we do not imply that there are direct interactions between EcR and the identified partner protein (lines 363-366). As for further biochemical and functional confirmation of interactions, we also agree with this. Unfortunately, at the moment we do not have antibodies to any of the discovered partners, and their generation will certainly take time. That is why we believe that this is a matter for future research. 

Reviewer 2 Report

Comments and Suggestions for Authors

This study evaluated an optimized proximity biotinylation workflow for Drosophila melanogaster tissues by comparing conventional streptavidin-agarose purification with two alternative approaches: anti-biotin antibodies and protease-resistant streptavidin (prS). The authors demonstrate that while antibody-based purification reduces background noise, it shows lower efficiency. Notably, they establish that prS maintains the high capture efficiency of traditional streptavidin while significantly minimizing nonspecific peptide interference in mass spectrometry analysis. The improved protocol was successfully applied to identify novel interaction partners of the ecdysone receptor (EcR) in Drosophila ovarian tissue using TurboID-mediated biotinylation, overcoming previous limitations in studying small tissue samples.

The proposed methodology has the potential to be valuable for future studies focusing on protein-protein interactions in a limited tissue sample. However, the manuscript requires substantial revisions to improve clarity, organization, and overall presentation of the data.

Major Revisions

A complete revision of both the Methods and Results sections is needed to ensure proper organization and clearer separation between methodological descriptions and experimental outcomes. In the text, the frequent inclusion of methodological justifications (e.g., the rationale for the PB approach, TurboID selection, and choice of EcR) within the Results section disrupts the manuscript’s logical flow and makes it difficult to follow.

I recommend restructuring the manuscript by summarizing the workflow in a schematic figure and moving the detailed protocols to the Supplementary Material. It is important to emphasize that the level of detail in the methods must be sufficient to ensure full reproducibility by other researchers, even if this requires extensive supplementary content. The main text should retain only the essential methodological information needed to support the narrative.

Additionally, the Discussion section currently deviates from the methodological focus of the study by engaging in speculative and unsubstantiated interpretations of EcR-related signaling pathways. To maintain alignment with the stated aim of optimizing proximity biotinylation workflows, I strongly recommend removing these speculative sections and refocusing the discussion on method validation, comparative efficiency, and applicability to small tissue samples. Alternatively, if the biological findings are to remain a central point, the Introduction should be revised to clearly state that EcR biology is also a major objective of the study.

Specific Revisions

Lines 103–105: While the authors cite a previously published protocol for protease-resistant streptavidin (prS) production (reference [8]), it is important to include at least a concise summary of the key steps within the manuscript. Given that prS is a central methodological innovation of the study, a brief protocol overview here should significantly enhance both reproducibility and reader understanding.

Lines 243–249: The detailed description of sample preparation (number of ovaries, biological replicates, and fractionation steps for Western blot) is more appropriate for the Methods section. This content should be relocated, while keeping in the Results section only the direct data interpretations (e.g., prS efficiency outcomes).

Lines 236–237: The explanation of control constructs (TurboID-Gly) and plasmid details is already covered in the Methods and could be removed from the Results. In this section, it is sufficient to refer to the controls when presenting the comparative results.

Lines 206–216: This paragraph includes several methodological and conceptual justifications (PB approach selection, TurboID choice, EcR rationale) that are not appropriate for the Results section. These points should be present in the Introduction or Methods, where the study rationale and experimental design choices are discussed.

Lines 364–368: This paragraph presents personal motivations ("our personal interest") and experimental rationale that would be more appropriately placed in the Introduction or Methods section. Additionally, this is too informal for a scientific manuscript. I recommend removing or rephrasing this section to use neutral, objective language, and relocating the relevant content to the appropriate section if necessary.

Comments on the Quality of English Language

In my opinion, English is sufficient for the research to be understood.

Author Response

This study evaluated an optimized proximity biotinylation workflow for Drosophila melanogaster tissues by comparing conventional streptavidin-agarose purification with two alternative approaches: anti-biotin antibodies and protease-resistant streptavidin (prS). The authors demonstrate that while antibody-based purification reduces background noise, it shows lower efficiency. Notably, they establish that prS maintains the high capture efficiency of traditional streptavidin while significantly minimizing nonspecific peptide interference in mass spectrometry analysis. The improved protocol was successfully applied to identify novel interaction partners of the ecdysone receptor (EcR) in Drosophila ovarian tissue using TurboID-mediated biotinylation, overcoming previous limitations in studying small tissue samples.

The proposed methodology has the potential to be valuable for future studies focusing on protein-protein interactions in a limited tissue sample. However, the manuscript requires substantial revisions to improve clarity, organization, and overall presentation of the data.

We are grateful to the Reviewer for his/her positive evaluation of our manuscript and for his/her opinion that it may be of interest to the public.

Major Revisions

A complete revision of both the Methods and Results sections is needed to ensure proper organization and clearer separation between methodological descriptions and experimental outcomes. In the text, the frequent inclusion of methodological justifications (e.g., the rationale for the PB approach, TurboID selection, and choice of EcR) within the Results section disrupts the manuscript’s logical flow and makes it difficult to follow.

We are grateful to the Reviewer for this valuable comment. We have substantially restructured the manuscript, moving the methodological aspect to the Methods and Supplementary Methods sections. We agree that this improved the logical flow of the manuscript.

I recommend restructuring the manuscript by summarizing the workflow in a schematic figure and moving the detailed protocols to the Supplementary Material. It is important to emphasize that the level of detail in the methods must be sufficient to ensure full reproducibility by other researchers, even if this requires extensive supplementary content. The main text should retain only the essential methodological information needed to support the narrative.

As suggested, we created two schemes describing experimental design and provided them as Fig.9 and Fig.10 (lines 379-385, 418-423). All information related to the experimental design and presented in the Results section of the previous version of the manuscript has now been moved to the Methods (brief summary) and Supplementary Methods (detailed description of the protocols) sections.

Additionally, the Discussion section currently deviates from the methodological focus of the study by engaging in speculative and unsubstantiated interpretations of EcR-related signaling pathways. To maintain alignment with the stated aim of optimizing proximity biotinylation workflows, I strongly recommend removing these speculative sections and refocusing the discussion on method validation, comparative efficiency, and applicability to small tissue samples. Alternatively, if the biological findings are to remain a central point, the Introduction should be revised to clearly state that EcR biology is also a major objective of the study.

We completely re-wrote the Discussion section. The main part of this section is devoted to the discussion of different approaches to performing the steps of the PB protocol in the case of small tissue samples (lines 250-322). However, we have retained a significantly shorter version of the discussion related to the identified EcR partners. We did it for two reasons:(1) indeed, the identified EcR partners are of some fundamental interest and are important for researchers studying the mechanism of transcriptional regulation by 20E; (2) during the revision, we received a separate request from the Editor to insert a paragraph into the Discussion that would place our research in the context of modeling human diseases in Drosophila.

Specific Revisions

Lines 103–105: While the authors cite a previously published protocol for protease-resistant streptavidin (prS) production (reference [8]), it is important to include at least a concise summary of the key steps within the manuscript. Given that prS is a central methodological innovation of the study, a brief protocol overview here should significantly enhance both reproducibility and reader understanding.

We put the detailed protocol into the Supplementary methods

Lines 243–249: The detailed description of sample preparation (number of ovaries, biological replicates, and fractionation steps for Western blot) is more appropriate for the Methods section. This content should be relocated, while keeping in the Results section only the direct data interpretations (e.g., prS efficiency outcomes).

We relocated the description (lines 399-407) and also provided it on the experimental scheme (lines 418-423).

Lines 236–237: The explanation of control constructs (TurboID-Gly) and plasmid details is already covered in the Methods and could be removed from the Results. In this section, it is sufficient to refer to the controls when presenting the comparative results.

We removed constructs description from the results (lines 399-403).

Lines 206–216: This paragraph includes several methodological and conceptual justifications (PB approach selection, TurboID choice, EcR rationale) that are not appropriate for the Results section. These points should be present in the Introduction or Methods, where the study rationale and experimental design choices are discussed.

We have removed this discussion from the Results section. Part of it has been moved to the Introduction and part of it to the Methods section.

Lines 364–368: This paragraph presents personal motivations ("our personal interest") and experimental rationale that would be more appropriately placed in the Introduction or Methods section. Additionally, this is too informal for a scientific manuscript. I recommend removing or rephrasing this section to use neutral, objective language, and relocating the relevant content to the appropriate section if necessary.

We apologize for the informal language. We tried to rephrase the manuscript and remove personal motivations.